# Socio-Economic, Demographic and Health Determinants of the COVID-19 Outbreak

**DOI:** 10.3390/healthcare10040748

**Published:** 2022-04-18

**Authors:** Ayfer Ozyilmaz, Yuksel Bayraktar, Metin Toprak, Esme Isik, Tuncay Guloglu, Serdar Aydin, Mehmet Firat Olgun, Mustafa Younis

**Affiliations:** 1Department of Foreign Trade, Kocaeli University, Kocaeli 41650, Turkey; ayfer.ozyilmaz@kocaeli.edu.tr; 2Department of Economics, Istanbul University, Istanbul 34452, Turkey; ybayraktar@istanbul.edu.tr; 3Department of Economics, Istanbul Sabahattin Zaim University, Istanbul 34303, Turkey; metin.toprak@izu.edu.tr; 4Department of Optician, Malatya Turgut Ozal University, Malatya 44700, Turkey; esme.isik@ozal.edu.tr; 5Department of Labor Economics and Industrial Relations, Yalova University, Yalova 77100, Turkey; tguloglu96@yahoo.com; 6School of Health Sciences, Southern Illinois University Carbondale, 1365 Douglas, Drive, Carbondale, IL 62901, USA; 7Rectorate, Kastamonu University, Kastamonu 37150, Turkey; mehmetfiratolgun@gmail.com; 8College of Health Sciences, Jackson State University, Jackson, MS 39217, USA; younis99@gmail.com

**Keywords:** COVID-19, prevalence of COVID-19, health, socioeconomic, quantile regression

## Abstract

Objective: In this study, the effects of social and health indicators affecting the number of cases and deaths of the COVID-19 pandemic were examined. For the determinants of the number of cases and deaths, four models consisting of social and health indicators were created. Methods: In this quantitative research, 93 countries in the model were used to obtain determinants of the confirmed cases and determinants of the COVID-19 fatalities. Results: The results obtained from Model I, in which the number of cases was examined with social indicators, showed that the number of tourists, the population between the ages of 15 and 64, and institutionalization had a positive effect on the number of cases. The results obtained from the health indicators of the number of cases show that cigarette consumption affects the number of cases positively in the 50th quantile, the death rate under the age of five affects the number of cases negatively in all quantiles, and vaccination positively affects the number of cases in 25th and 75th quantile values. Findings from social indicators of the number of COVID-19 deaths show that life expectancy negatively affects the number of deaths in the 25th and 50th quantiles. The population over the age of 65 and CO_2_ positively affect the number of deaths at the 25th, 50th, and 75th quantiles. There is a non-linear relationship between the number of cases and the number of deaths at the 50th and 75th quantile values. An increase in the number of cases increases the number of deaths to the turning point; after the turning point, an increase in the number of cases decreases the death rate. Herd immunity has an important role in obtaining this finding. As a health indicator, it was seen that the number of cases positively affected the number of deaths in the 50th and 75th quantile values and the vaccination rate in the 25th and 75th quantile values. Diabetes affects the number of deaths positively in the 75th quantile. Conclusion: The population aged 15–64 has a strong impact on COVID-19 cases, but in COVID-19 deaths, life expectancy is a strong variable. On the other hand, it has been found that vaccination and the number of cases interaction term has an effect on the mortality rate. The number of cases has a non-linear effect on the number of deaths.

## 1. Introduction

The World Health Organization (WHO) China Country Office reported pneumonia (pneumonia) cases of an unknown etiology in Wuhan, China, on 31 December 2019. On 7 January 2020, this type of virus was identified as a new Coronavirus (2019-nCoV) that has not been previously detected in humans. Later, the name of the 2019-nCoV disease was accepted as COVID-19, and the virus was even named SARS-CoV-2 because of its close resemblance to SARS CoV. Coronavirus belongs to the same family of coronaviruses that caused the 2003 Severe Acute Respiratory Syndrome (SARS) and 2012 Middle East Respiratory Syndrome (MERS) outbreak. As a result of research, it has been revealed that SARS-CoV is infected by musk cats, and MERS-CoV is transmitted from single-humped camels. SARS-CoV emerged as a previously unknown virus in 2003 as the first international health emergency of the 21st century, causing hundreds of people to die. About 10 years later, MERS-CoV (Middle East Respiratory Syndrome Coronavirus) from the Coronavirus family, which has not previously been demonstrated in humans or animals, was first described in humans in Saudi Arabia in September 2012; however, it was later revealed that the first cases were seen in a hospital in Zarqa, Jordan, in April 2012. While the mortality rate is around 10% for SARS and 34% for MERS, this rate for COVID-19 is about 1% to 3.4%. However, when compared with seasonal influenza, with a mortality rate of less than 1%, the mortality rate of coronavirus appears to be significantly higher [1,2,3]. However, the most prominent feature of coronavirus is its threat to society, in that it spreads very rapidly, collapses the health infrastructure, and causes social panic. In this aspect, it differs from the others.

The virus is economically contagious as well as medically. Almost all countries integrated into the world trade system are affected by disruptions in the supply chain. The crisis caused by the outbreak may end sharply and sharply with the discovery of treatment; however, it also has the possibility to extend over the longer term. Therefore, it is difficult to predict the magnitude and continuity of the economic impact caused by the outbreak [4]. Although COVID-19, which affects almost all sectors, may end soon, it is expected that the world economy will take a long time to recover [5] because the pandemic has affected world trade [6] and the supply chain.

Two main policies are vital in pandemic processes: First, to prevent the spread by controlling the outbreak. Second, to find a way through the process with the least amount of damage without shutting down the engine of the economy. For this, policymakers focus on different policy combinations by considering the structural conditions of their economies. Supporting credit institutions, debt deferral and low-interest rates for companies, unemployment benefits, national support campaigns for the needy, premium payments to healthcare workers, cash support provided to citizens, follow-up and penalties especially for medical supplies hoarders, stock management for basic food and medical supplies, the prevention of news and social media sharing that led to public panic are some of the measures taken.

Although this epidemic affects the entire society, it causes disproportionately more damage to the poor, unemployed, refugees, and individuals working in the essential sectors for low wages [7]. In this process, the lives of refugees, who are seen as a burden due to security [8,9] and economic effects [10,11] in the host countries, are affected much more deeply than the local people.

Determining the factors affecting COVID-19 is very important in the fight against the epidemic [12]. In this context, which factors are decisive in the spread of COVID-19 or increasing the number of deaths is one of the issues frequently discussed in the epidemic literature. In this context, some studies emphasize health indicators, while others emphasize socio-economic factors. Indicators such as demographic factors, chronic diseases, socio-economic variables, and health systems are frequently discussed topics within the scope of determinants of COVID-19 [13,14,15].

Since the early days of the epidemic, many studies have been conducted discussing the determinants of COVID-19. In this context, Koc and Sarac [16] discussed the relationship between the number of COVID-19 cases, deaths and death rates, and indicators such as socio-economic, demographic, and health in OECD countries. According to the findings, there is a positive correlation between the share of health expenditures in GDP and COVID-19 cases and deaths. The most important explanation for this may be the inadequate measures taken by OECD countries, which have a high share of health expenditures in GNP, at the beginning of the epidemic. In addition, the increase in the prevalence of chronic diseases such as obesity, diabetes, and smoking increase the number of COVID-19 cases and deaths. The factors that affect the COVID-19 death rate are the severity index, life expectancy at birth, smoking, and the share of health expenditures in GDP. Focusing on a similar relationship, Ehlert [17] found that there is a positive relationship between the number of COVID-19 cases and the average age, population density, and the share of people employed in aged care in Germany. However, it has been suggested that there is a negative relationship between COVID-19 and school children and the density of doctors. According to Dintrans et al. [18], the poverty index, use of public transport, population over 65, population density, rurality, self-employment, green spaces, and difficulty in finding health care are the determining factors of COVID-19 in Chile. Rahman et al. [12] discussed potential factors affecting the incidence rates of COVID-19 in Bangladesh. According to the findings of the study, there is a statistically significant relationship between incidence rates of COVID-19 and the percentage of the urban population, the number of health workers, and the distance to the capital city. Khobragade and Kadam [19] suggested that states with high literacy rates in India have a lower COVID-19 death rate. In addition, urban population, migration, and per capita health expenditures increase COVID-19 deaths, but population density is not an important determinant of COVID-19. On the other hand, Parohan et al. [20] conducted a meta-analysis of studies discussing risk factors for mortality in patients with COVID-19. The meta-analysis included 14 studies involving 29,909 COVID-19 infected patients and 1445 deaths. Accordingly, there is a positive relationship between aged 65 and <65 years, male gender, hypertension, cardiovascular diseases, diabetes, COPD and cancer, and COVID-19 deaths. Chung et al. [21] focused on the relationship between the income-poverty rate and COVID-19 in Hong Kong. According to the study, there is no significant relationship between these variables. Bayraktar et al. [22] focused on the role of the health system in the fight against COVID-19 in 124 countries. According to the study findings, the number of doctors, the number of hospital beds, and the life expectancy at the age of 60 have a positive effect on the COVID-19 recovery/case rate, but the share of health expenditures in GDP and universal health service is not statistically significant.

On the other hand, Perone [23] discussed the reasons for differences in COVID-19 case fatality rates with a large heterogeneous group of factors. According to the study, in general environmental, demographic, and health-related factors play an important role in the explanation of COVID-19 case deaths in 20 regions and 107 provinces in Italy. In this context, overall health care efficiency, doctor density, and average temperature decrease the case fatality rate. On the other hand, the population aged 70 and over, the density of cars and companies, air pollution, and relative average humidity increase the COVID-19 case fatality rate. One of the studies discussing the determinants of COVID-19 cases and deaths with a large variable dataset belongs to Bhowmik et al. [24]. In the study for the USA, the determining factors of COVID-19 were categorized as socio-demographic characteristics, health indicators, mobility trends, and health services infrastructure. According to the study, the increase in the rate of women and the young population (<18 years old), the increase in the share of individuals with less than high school education, and the increase in employment increase the incidence of COVID-19. In contrast, the percentage of people living in rural areas is negatively correlated with COVID-19 cases. Because rural areas are not densely populated, therefore, these places are more suitable for social distance. According to the study, which also emphasizes racial differences, counties with a higher proportion of African-Americans have a higher COVID-19 death rate. In addition, counties with higher income inequality rates have higher COVID-19 deaths. When health indicators are examined, more COVID-19 deaths are seen in districts with a higher number of HIV, cancer, hepatitis-A, and cardiovascular patients. Because these diseases generally weaken immunity and make individuals more vulnerable to diseases. Another finding of the study is that the number of intensive care beds per capita reduces the COVID-19 mortality rate. Focusing on similar variables, Xie and Li [25] examined the relationship between COVID-19 cases and deaths, health, and demographics in US states. According to the study, the rate of COVID-19 infection is positively correlated with both the percentage of adults with high school education or less and population density. However, the rate of COVID-19 infection is negatively correlated with the percentage of smokers, the percentage of unsafe food, the percentage of adults with obesity, and the dissociation index. On the other hand, the COVID-19 mortality rate is positively correlated with the elderly population aged 65 and over and the rural population and population density. According to the study, which also includes vaccination, there is a negative relationship between the rate of COVID-19 infection and the percentage of vaccination.

Unlike these studies, Dalsania et al. [26] discussed the relationship between social determinants of health and COVID-19 mortality, focusing on racial differences in mortality in the USA. According to the study, COVID-19 death rates were higher in areas with dense Black populations. In this context, the increase in the percentage of Black residents, percent low birthweight, households without internet, uninsured adults, and the percentage of adults without a high school diploma increase COVID-19 death rates. One of the studies focusing on racial factors is the study of McLaughlin et al. [27] for the USA. Accordingly, the ratio of the racial/ethnic minority population, income inequality, uninsured people, air pollution, and diabetes increase the COVID-19 case and death rates. One of the studies emphasizing different parameters is the study of Amaratunga et al. [28]. In addition to demographic factors, the study focused on geographical variables such as access to food and health facilities. Accordingly, in most models, there are differences between the frequency of McDonald’s and the frequency of other fast-food or non-fast-food restaurants. McDonald’s frequency is associated with fewer cases (deaths), but other fast-food or non-fast food restaurants are associated with more cases (deaths). The reason for this is that McDonald’s attaches importance to hygiene or is meticulous about control measures for crowding. On the other hand, pharmacy frequency reduces COVID-19 deaths, but gyms and low-income percentage increase cases and deaths. Magalhães et al. [29] examined the relationship between the risk of COVID-19 infection and socioeconomic deprivation. According to the findings of the study for Portugal, socioeconomic deprivation is effective on the risk of SARS-CoV-2 infection. Because the most deprived population work in basic sectors and have worse housing conditions. Maiti et al. [30] focused on crime, income, and immigration in the USA. According to the study, there is a strong correlation between these variables and COVID-19. According to the study, which also includes climate and air pollution, there is no significant relationship between these variables and COVID-19 cases or deaths. Palacio and Tamariz [31], in their study on the importance of stress, suggested that stress is associated with COVID-19 infection in South Florida.

Some studies focused more on demographic variables. For example, Sannigrahi et al. [14] discussed the impact of demographic parameters on COVID-19 cases and deaths in Europe. Wide demographic data were used, such as long-term illness, age >80, employment, life expectancy, crude birth rate, the population aged 15–24, the population aged 50–65, the population aged 65–80, the population aged 0–14, the population aged 25–49 and, in general, it has been found that these factors are effective on COVID-19. Similarly, Drefahl et al. [32] suggested that the male population, less disposable income, lower education level, being unmarried, and being a migrant from a low- or middle-income country increases the risk of death from COVID-19 in Sweden. The role of socio-economic characteristics is more prominent in working-age groups, while the role of marital status is more prominent in retirement-age groups. In summary, COVID-19 creates an unequal burden on the disadvantaged and vulnerable members of society.

The prominent parameter among the demographic variables is undoubtedly the population density. There are many studies in the literature focusing on population density. For example, Jo et al. [33] found that population density is statistically significant in explaining the spread of COVID-19 in Korea. Xie and Li [24] found that there is a positive correlation between population density and the rate of COVID-19 infection in US counties. Arbel et al. [34] suggested that population density increases the probability of COVID-19 infection in Israel. Wong and Li [35] revealed that population density is an important parameter in the case increase in COVID-19 in the USA. However, Sun et al. [36] suggested that population density was not an important factor in the spread of COVID-19 under strict quarantine policies in China. According to the study, the main reason for this is that China’s quarantine policies can effectively control the spread of COVID-19. In addition, Palacio and Tamariz [31] in Miami-Dade County, and Khobragade and Kadam [19] in India, suggested that population density is not a predictor of COVID-19.

The main solution is to find treatment methods that will reduce the spread of the outbreak and ultimately end it. Therefore, this study is important in terms of revealing which variables affect the rates of cases and fatalities in countries with low and high concentrations of cases and case-related deaths. In this context, considering the variables related to the COVID-19 outbreak, which are thought to be effective on the number of cases and deaths, the extent to which these variables affect the outbreak was analyzed. In the study, the cross-sectional regression method was used.

## 2. Material and Methods

In this study, 4 different models are defined. Model I and Model II include socio-economic indicators and health indicators that are effective in COVID-19 cases, respectively. Model III and Model IV include socio-economic indicators and health indicators that are effective in COVID-19 deaths, respectively. Due to the availability of data, the country sample in all models is limited to 93 countries. The 6 March 2022 data were used in case and vaccination rates. In other variables, the years obtained according to the accessibility of the data were taken, and the latest 2017 data were used. The cross-sectional equations for Model I, Model II, Model III, and Model IV are as follows:

Model I (Case Model with Social Indicators):(1)$$\mathrm{LCasePer}=\beta_{0}+\beta_{1}\mathrm{LTourist}+\beta_{2}\mathrm{LPopDensity}+\beta_{3}\mathrm{LUrban}+\beta_{4}\mathrm{LPop}+\beta_{5}\mathrm{LDI}+u\;$$

Model II (Case Model with Health Indicators):(2)$$\mathrm{LCasePer}=\beta_{0}+\beta_{1}\mathrm{LSmoking}+\beta_{2}\mathrm{LLife}+\beta_{3}\mathrm{LNursing}+\beta_{4}\mathrm{LVacper}100+\beta_{5}{\mathrm{LVacper}}^{2}+u$$

Model III (Death Model with Social Indicators):(3)$$\mathrm{LTotDeath}=\beta_{0}+\beta_{1}\mathrm{LLifeEx}+\beta_{2}\mathrm{LPop}65+\beta_{3}\mathrm{LCHE}+\beta_{4}{\mathrm{LCO}}_{2}+\beta_{5}\mathrm{LCasePer}+\beta_{6}{\mathrm{LCasePer}}^{2}+u$$

Model IV (Death Model with Health Indicators):(4)$$\mathrm{LDeathPer}=\beta_{0}+\beta_{1}\mathrm{LCasePer}+\beta_{2}\mathrm{LFullVac}+\beta_{3}\mathrm{LCasePer}\times \mathrm{LFullVac}+\beta_{4}\mathrm{LDiabet}+\beta_{5}\mathrm{LDoctor}+\beta_{6}\mathrm{LCHE}+u\;$$

The interaction variable was used for Model IV. This variable is LCasePer X LFullVac. The total effect of COVID-19 cases (per million people) on COVID-19 deaths (per million people) given fully COVID-19 vaccineted persons are computed as:(5)$$\frac{\partial \mathrm{LDeathPer}}{\partial \mathrm{LCasePer}}=\beta_{1}+\beta_{3}\mathrm{LFullVac}\;$$

The total effect of LFullVac on COVID-19 deaths (per million people) given COVID-19 cases (per million people) are computed as:(6)$$\frac{\partial \mathrm{LDeathPer}}{\partial \mathrm{LFullVac}}=\beta_{2}+\beta_{3}\mathrm{LCasePer}\;$$

The variables and data sources used in the models are presented in Table 1.

In this study, the quantile regression method developed by Koenker and Bassett [37] was used. Quantile Regression allows estimating parameters in different quantile values of the conditional distribution of the dependent variable and provides robust estimation results in the presence of excessive values. This method does not assume that the error term is normally distributed. The minimization problem that needs to be solved for θ quantile regression is as follows:(7)$$\underset{\beta \in R^{K}}{\min}\left[\underset{i\in \left\{i:{\;y}_{i}\geq x_{i}^{\prime }\beta \right\}}{\sum }\theta \left|y_{i}-x_{i}^{\prime }\beta \right|+\underset{i\in \left\{i:{\;y}_{i}<x_{i}^{\prime }\beta \right\}}{\sum }\left(1-\theta \right)\left|y_{i}-x_{i}^{\prime }\beta \right|\right]$$

Here θ shows the quantile levels and takes values between 0 and 1. $y_{i}$ is the dependent variable and $x_{i}$ is the vector of explanatory variables in the dimension Kx1 [17]. The conditional quantile of $y_{i}$ given $x_{i}$ is:(8)$$Q_{y}\left(\theta |x_{i}\right)=x_{i}^{\prime }\beta_{\theta }$$
here, the slope parameters can be estimated for the desired quantile (θ) level [38,39,40].

## 3. Results

In this study, the determining variables for the number of COVID-19 cases were estimated by Model I and Model II, and the determining variables for the number of COVID-19 deaths were estimated by Model III and Model IV. The quantile regression method was used in these models estimations. In the estimates, 25th, 50th, and 75th quantiles were used, and the estimation results are presented in Table 2.

When the Model I estimation results for the determinants of the number of COVID-19 cases social indicators are analyzed, it is seen in LTourist, which shows international tourism, that the number of arrivals has a statistically significant and positive effect on COVID-19 cases in the 25th and 50th quantiles. In other words, as the number of tourist arrivals increases, the number of cases increases. Namely, in countries where the number of cases is low (25th quantile), a 1% increase in tourist numbers increases the average number of cases per million people by 0.2797%; in countries with a medium number of cases (50th quantile), a 1% increase in tourist number increases the average number of cases by 0.173%. The LUrban 50th and 75th quantile values, which show the urban population (% of total population), are statistically significant. In countries where cases are high (75th quantile), a 1% increase in the urban population increases the number of cases per 100,000 people by 0.954%. LPop and LDI have a statistically significant effect on the case in all quantile values. It is seen that the population between the ages of 15-64 has a very important effect on the increase in the number of cases, as they are more actively involved in social life. If the share of the population aged 15–64 in the total population increases by 1%, the number of cases per million people increases by 9.975% in the 25th quantile. In other quantile values, this effect has a very high degree. Population density, on the other hand, does not show a statistically significant effect on the number of cases in all quantile values. The findings of Sun et al. [36] and Palacio and Tamariz [31] on some states in the USA also show that population density does not affect the number of cases.

Considering the results of Model II, where the number of cases is handled with health indicators, mortality rate, under-5 (per 1000 live births) has a statistically significant effect on the number of cases in all quantile values. In countries with a low number of cases (25th quantile), a 1% increase in the mortality rate under-5 (per 1000 live births) reduces the number of cases by 1.057%. As the vaccination rate increases, the number of cases is positively affected. The main reason for this is that vaccinated people behave more comfortably and continue their social lives actively despite the pandemic. As vaccination (per 100,000 people) increases by 1%, the number of cases (per million people) increases by 2.82% in the 25th quantile. In countries where the number of cases is high, this effect is higher. Namely, in countries where the number of cases (per million people) is high (75th), if the number of vaccinations increases by 1%, the number of cases increases by 1.86%. Considering that the effect of the vaccination rate on the number of cases may be non-linear, the square of the vaccination rate (per 100,000 people) was also included in the model. However, the square of the vaccination rate was found to be statistically insignificant. Therefore, it can be said that the effect of vaccination on the case is linear. Cigarette consumption is significant only to the 50th quantile. A 1% increase in cigarette consumption increases the number of cases by 0.438%.

When the Model III estimation results, which include the main social variables to explain the deaths due to COVID-19, are considered, the number of cases (per million) has a non-linear effect on the number of deaths. Case number and square are statistically significant in the number of deaths in the 50th and 75th quantiles. Therefore, the number of cases has a non-linear effect on the number of deaths. The fact that the positive square of the number of cases is negative indicates that the number of cases reduces the number of deaths after a certain level. Very high case numbers reduce mortality rates through the acquisition of herd immunity. Therefore, the increase in the number of cases up to the turning point (58,562) for the 50th quantile increases the death rates, while the increase in the number of cases after this point decreases the death rates, which is in line with the expectations. In the 75th quantile, the turning point was calculated as 92,277. Life expectancy at birth (LLife) and the share of the population over the age of 65 in the total population (LPop65) have a significant effect on the number of deaths. Life expectancy at birth has a very strong effect on the number of deaths from COVID-19 in countries with a low number of cases (25th). A 1% increase in life expectancy reduces the number of deaths due to COVID-19 by 10.82%. The population over the age of 65 has a statistically significant effect on the number of COVID-19 deaths in all quantile values. If the share of the population over the age of 65 in the total population increases by 1% at the 25th, 50th, and 75th quantile values, the number of deaths due to COVID-19 increases by 0.9313%, 1.0609%, and 0.6875%. Estimation results show that the elderly population has an excessive effect on the COVID-19 death number. The main reason for this is that the elderly are in the high-risk group for COVID-19. Another variable that affects the number of deaths in all quantile values is the CO_2_ emission. There is a positive relationship between carbon emissions and the number of deaths. A 1% increase in carbon dioxide emissions increases the number of deaths by 0.9158% in the 25th quantile. The variables used in the model showed results consistent with expectations.

When we look at the results of the model (Model IV), in which the number of deaths due to COVID-19 is handled with health indicators, it is seen that the number of cases (per million people) is the main determinant of the number of deaths. The number of cases has a statistically significant effect on all quantile values. The variable, which is the interaction of the number of cases and the vaccinated person, is statistically significant in the 25th and 75th quantiles. The impact on mortality will be different for vaccinated and COVID-19-infected individuals. Therefore, the interaction of the number of cases (per 100,000 people) and the fully vaccinated (percentage of population) was added to the model as a variable in this model. The net effect of the number of cases on the number of deaths from COVID-19 is 0.6853 (0.645 + 0.0418). In other words, a 1% increase in the number of cases increases the number of deaths by 0.6853%. The net effect of the number of vaccinations on the case is −0.7024. A 1% increase in the fully vaccinated reduces the number of deaths by 0.7024%. LDiabet variable, which shows diabetes prevalence (% of the population age 20 to 79), has a statistically significant and positive effect on the number of deaths (per person) only in the 75th quantile. A 1% increase in diabetes prevalence increases the number of deaths by 0.456%. This indicates that diabetic patients are significantly affected, especially in countries with high deaths, and that diabetes patients are among high-risk groups.

The coefficients of the variables that affect the number of COVID-19 cases and deaths follow different trends in different quantile values. These differences are illustrated in the graphics below. For Model I in Figure 1, Model II in Figure 2, Model III in Figure 3 and Model IV in Figure 4, the coefficients of different quantiles remain within the confidence interval; therefore, the graphs show that the estimating results are consistent and reliable. The graphical representation of the quantile regression results showing the slope of each variable coefficients at different quantile values is as follows:

## 4. Conclusions

This study aimed to obtain the factors determining the spread of COVID-19 cases and deaths with a cross-sectional quantile regression method. This method is important in terms of revealing which variable is effective in the countries where the density of the number of cases and the number of deaths are low and high. In this framework, the data of 93 countries are used. In Model I, the effect of social indicators on COVID-19 cases was examined. In the second model, the effect of health indicators on the number of cases is discussed. On the other hand, two models have been created to examine deaths due to COVID-19 with social and health indicators. Model III, where the number of deaths is handled with social indicators. In Model IV, COVID-19 deaths were discussed with health indicators.

The explanatory variables for the number of cases with social indicators, number of tourist arrivals, population density, urban population (% of the total population), population ages 15–64 (% of the total population), and democracy were used. For the health indicators of the number of cases, which is another model, the explanatory variables are the prevalence of current tobacco use (% of adults), mortality rate, under-5 (per 1000 live births), nursing personnel, and COVID-19 vaccine doses administered (per 100,000 people) and the square of COVID-19 vaccine doses administered. In Model III, in which the number of deaths due to COVID-19 is handled with social indicators, the explanatory variables are life expectancy at birth, the population aged 65 and above, domestic private health expenditure (% of the current health expenditure), CO_2_ emissions (kt), COVID-19 cases per million people and the square are used. In Model IV, in which the number of COVID-19 deaths is discussed with health indicators, COVID-19 cases (per million people), the population fully vaccinated against COVID-19, interacting terms for case and vaccination, diabetes prevalence, the density of medical doctors, and current health expenditure variables (% of GDP) are included in the model.

In this study, it was seen that the population between the ages of 15 and 64, which is one of the social indicators, is highly effective in representing the number of cases. The dominant effect of the group aged 15–64 is valid in all countries where the number of cases is low, medium, and high. The global freedom score (LDI) is the second variable that has a prominent effect on the number of cases. Institutionalization positively affects the number of cases at all quantiles. In countries where the number of cases is low, this effect is quite high, while the effect decreases as the number of cases increases. In countries where the number of cases is high (75th quantile), urbanization is a factor that increases the number of cases. The urbanization rate in the country is more effective on the number of cases than the population density. The number of tourist arrivals to the country, which is another social variable, also causes an increase in the number of cases.

As an indicator of health, vaccination is the most important variable that increases the number of cases. It positively affects the number of cases in all quantiles. It was concluded that the relationship between the case and vaccination was linear. The second most important health indicator that affects the number of cases is the mortality rate, under-5 (per 1000 live births). In all countries where the number of cases is low, medium, and high, the mortality rate under-5 decreases the number of cases. Smoking, on the other hand, has an increasing effect on the number of cases in countries where the number of cases is average (50th quantile). The effect of smoking on the number of cases is relatively low.

The results of Model III, in which deaths related to COVID-19 are handled with social indicators, show that life expectancy is the most important indicator. Life expectancy affects the number of deaths with a very high coefficient. This effect is valid for 25th and 50th quantile values. COVID-19 mortality decreases as life expectancy increases. In countries where the number of deaths due to COVID-19 is low (25th quantile), a 1% increase in life expectancy reduces the number of deaths by 10.827%. The population aged 65 and above is highly reflected in the number of deaths. The dominant effect of the group aged 65+ is valid in all countries where the number of deaths is low, medium, and high. The population aged 65 and over is the highest risk group in the COVID-19 outbreak. The increase in the population aged 65 and over increases the number of deaths in line with expectations. Carbon emission has a statistically significant and positive effect on the number of COVID-19 deaths. The results show that CO_2_ emission is a critical factor likely to increase total coronavirus death rates. This result is in parallel with the results of Shamsi et al. [41]. The number of cases has a statistically strong and significant effect on the number of COVID-19 deaths. The quadratic inclusion of the number of cases in the model shows that there is a non-linear relationship between the number of cases and COVID-19 deaths. After the increase in the number of cases reaches a certain level, it has a negative effect on the number of deaths. Accordingly, if the number of cases exceeds a certain threshold, it leads to herd immunity, and accordingly, mortality rates decrease.

The results obtained from Model IV indicate that there are also non-linear effects on the number of COVID-19 deaths. It has been observed that the number of cases and vaccination together indirectly affect the number of deaths. In Model IV, where the number of cases and the rate of vaccination were included in the model as interaction variables, the final effect of the number of cases was estimated as 0.6853 and the final effect of vaccination as −0.7024 in the 25th quantile. A 1% increase in the number of cases increases 0.685% in the 25th quantile. A 1% increase in the vaccination rate reduces the number of deaths by 0.702% in the 25th quantile. In countries where the number of deaths from COVID-19 is high, the number of diabetes patients is reflected in the number of deaths due to COVID-19. A 1% increase in diabetes prevalence increases the death from COVID-19 by 0.456% in the 75th quantile. Kutlutürk [42] also indicates that individuals infected with COVID-19 have high mortality rates. The results show that diabetes patients should be more cautious and careful regarding COVID-19.

The following issues should be considered by policymakers in reducing the number of cases and controlling the outbreak:Since the population aged 65 and above is the variable that increases the number of cases and deaths, public regulations to prevent virus contact with this population should be strictly implemented.The relationship between diabetes and epidemic reveals that individuals with diabetes should be more careful in precautions such as social distance and hygiene.Trust in health policies should be ensured through institutionalization, transparency, and access to information.

## Figures and Tables

**Figure 1 healthcare-10-00748-f001:**
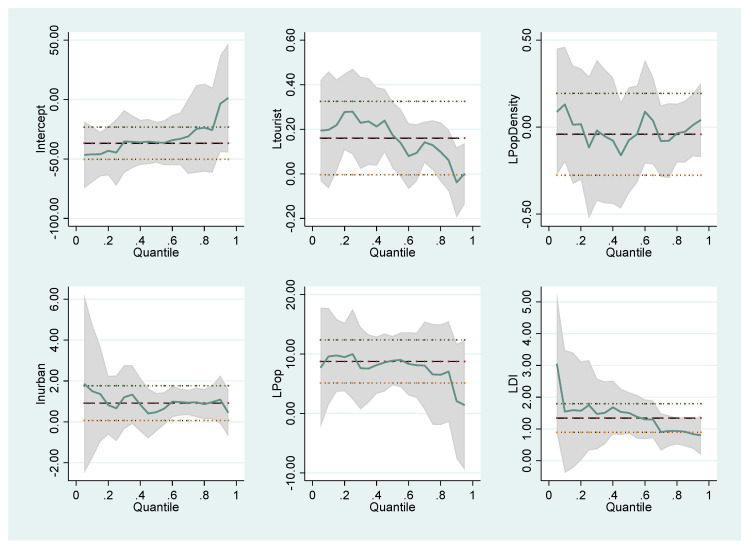
Changes in the quantile regression coefficients for Model I. Notes: The shaded bands represent the corresponding 95% confidence intervals.

**Figure 2 healthcare-10-00748-f002:**
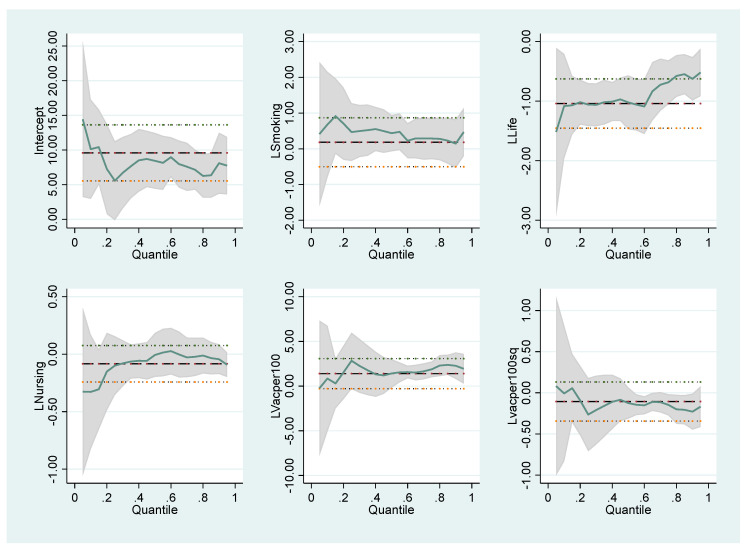
Changes in the quantile regression coefficients for Model II. Notes: The shaded bands represent the corresponding 95% confidence intervals.

**Figure 3 healthcare-10-00748-f003:**
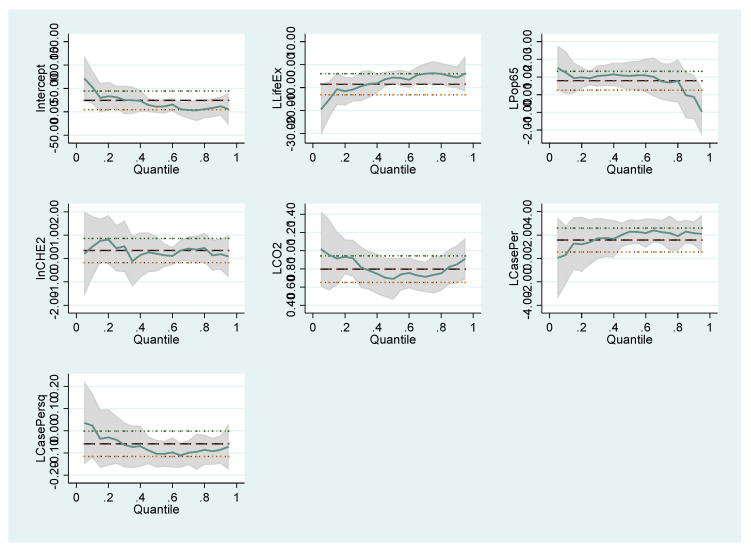
Changes in the quantile regression coefficients for Model III. Notes: The shaded bands represent the corresponding 95% confidence intervals.

**Figure 4 healthcare-10-00748-f004:**
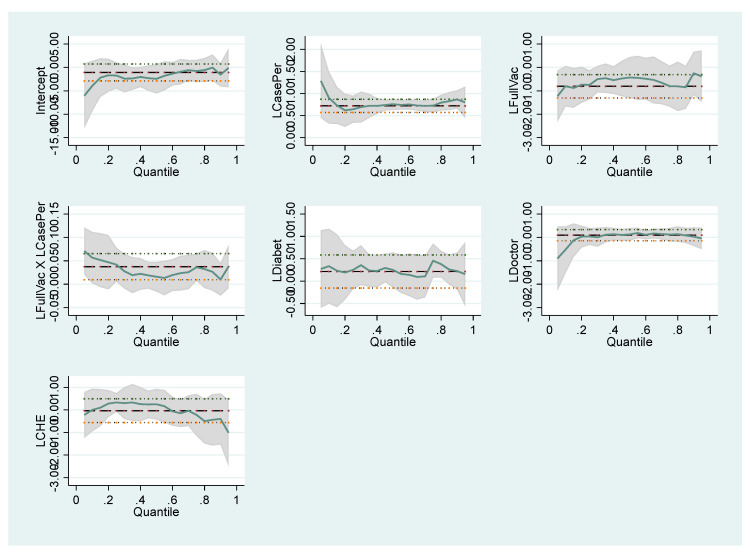
Changes in the quantile regression coefficients for Model IV. Notes: The shaded bands represent the corresponding 95% confidence intervals.

**Table 1 healthcare-10-00748-t001:** Data Description.

Variables	Description	Data Source
LTotDeath	Number of COVID-19 Deaths	Our Data in World
LCasePer	COVID-19 cases per million people	Our Data in World
LDeathPer	COVID-19 deaths per million people	Our Data in World
LTourist	International tourism, number of arrivals	World Bank Open Data
LPopDensity	Population density (people per sq. km of land area)	World Bank Open Data
LUrban	Urban population (% of total population)	World Bank Open Data
LPop	Population ages 15–64 (% of total population)	World Bank Open Data
LDI	Global Freedom Scores (Political Rights + Civil Liberties)	Freedom House
LSmoking	Prevalence of current tobacco use (% of adults)	Our Data in World
LLife	Mortality rate, under-5 (per 1000 live births)	World Bank Open Data
LNursing	Nursing personnel (number)	WHO
LVacper	COVID-19 vaccine doses administered per 100,000 people	Our Data in World
LLifeEx	Life expectancy at birth	World Bank Open Data
LPop65	Population ages 65 and above (% of the total population)	World Bank Open Data
LCHE	Domestic private health expenditure (% of current health expenditure)	World Bank Open Data
LCO_2_	CO_2_ emissions (kt)	World Bank Open Data
LFullVac	Fully vaccinated against COVID-19 (Percentage of population)	Our Data in World
LDiabet	Diabetes prevalence (% of population ages 20 to 79)	Our World in Data
LDoctor	Density of medical doctors (per 10,000 population)	WHO

Note: L shows the logarithmic transformation of all variables.

**Table 2 healthcare-10-00748-t002:** Quantile Regression Results.

Models and Variables	Q25	Q50	Q75
Model I (Case Model with Social Indicators)			
LTourist	0.2792 *** (0.1018)	0.1738 ** (0.0776)	0.1287 (0.0816)
LPopDensity	−0.1173 (0.1552)	−0.0748 (0.1744)	−0.0781 (0.1187)
LUrban	0.6618 (0.5819)	0.4719 *** (0.3964)	0.9545 *** (0.3825)
LPop	9.9750 *** (2.6028)	8.8773 *** (1.9249)	6.5700 ** (2.8073)
LDI	1.7466 *** (0.5651)	1.5028 *** (0.3251)	0.9379 *** (0.3160)
Model II (Case Model with Health Indicators)			
LSmoking	0.4691 (0.3842)	0.4384 ** (0.2083)	0.2885 (0.2883)
LLife	−1.0572 *** (0.1432)	−1.0196 *** (0.1635)	−0.6831 *** (0.1836)
LNursing	−0.0994 (0.0971)	−0.0069 (0.1039)	−0.0227 (0.0726)
LVacper	2.8253 ** (1.2422)	1.4093 (0.9509)	1.8695 ** (0.7774)
LVacper^2^	−0.2644 (0.1649)	−0.1242 (0.1226)	−0.1458 (0.0970)
Model III (Death Model with Social Indicators)			
LLifeEx	−10.8273 *** (2.6738)	−5.6642 *** (2.0035)	−3.7322 (3.2050)
LPop65	0.9313 ** (0.3948)	1.0609 ** (0.4564)	0.6875 * (0.3907)
LCHE	0.4401 (0.4602)	0.2062 (0.3313)	0.3797 * (0.2254)
LCO_2_	0.9158 *** (0.0960)	0.6895 *** (0.0959)	0.7330 *** (0.0888)
LCasePer	1.3796 (1.0602)	2.2790 *** (0.6390)	2.1699 *** (0.8066)
LCasePer^2^	−0.0411 (0.0593)	−0.1038 *** (0.0362)	−0.0949 * (0.0496)
Model IV (Death Model with Health Indicators)			
LCasePer	0.6435 *** (0.1358)	0.7578 *** (0.1081)	0.7266 *** (0.0708)
LFullVac	−0.7442 *** (0.2158)	−0.4312 (0.5259)	−0.8020 * (0.4227)
LDiabet	0.2499 (0.2826)	0.2493 (0.2002)	0.4565 ** (0.1887)
LCaseperXFullVac	0.0418 *** (0.0110)	0.0160 (0.0221)	0.0362 ** (0.0177)
LDoctor	0.0396 (0.2650)	0.1301 (0.2048)	0.1191 (0.1432)
LCHE	0.3331 (0.3472)	0.2528 (0.3380)	−0.2190 (0.7426)

Notes: The numbers in parentheses are heteroskedasticity robust standard errors. ***, ** and * indicate statistical significance at 99%, 95% and 90% significance levels.

## Data Availability

Not applicable.

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
