# Peer review of "Socio-Economic, Demographic and Health Determinants of the COVID-19 Outbreak"

_healthcare, 2022, doi:10.3390/healthcare10040748_

Round 1
Reviewer 1 Report
This paper is supposed to be devoted to quantile regression applied to COVID-19 prediction. The theme is interesting, but the manuscript in its present form is not focused to the main theme.
Argue on the novel contributions of the paper. What are the differences against previous publications co-authored by some of the authors of this paper?
Section 2 on ANNs is naive. It should be shortened. Figures 208 are irrelevant to the paper theme.
Is it R or R^2 in p.4?
Figures 9 and 10 should be described in detail.
Please explicit state whether an existing matlab function has been used or a new matlab function was been developed by the authors.
Streamline the cited references.
Author Response
Thank you for the valuable feedback. Each of the comments has been addressed and improved based on the request. Look forward to seeing it published. Best,
General Responses
We are grateful to receive the invaluable comments of the editor and the reviewers. We believe that the quality and comprehensibility of the manuscript have significantly improved after making the requested changes.
Throughout this document, we present a point-by-point reply to the reviewers’ concerns. For better readability, our revisions are shown in red. We thank you for your time and hope that you find our new revision satisfactory.
Response to Reviewer 2
- i) As per your suggestion, the study was expanded and focused on the main theme.
- ii) In our other published study, we focused on the health system and the impact of health system indicators on the number of COVID-19 recoveries/cases are discussed. In this study, a wide data set and socio-demographic and health indicators that have an effect on both the number of cases and the number of deaths were used. In the study, the number of models was increased to 4 by separating health indicators and socio-demographic indicators and increasing the number of variables.
iii) The interaction variable was used for Model 4. In this context, this interaction variable was added to the study, considering that the number of cases and the vaccine may have an effect on the number of Covid-19 deaths together.
iv)Since the ANN method was removed from the model, the title was revised accordingly.
Reviewer 2 Report
My main concern is about the method. On page 7, the two models only consider linear effects and there is no interaction. The authors need to explain why their final models do not include any possible non-linear effects or interactions among the variables.
Author Response
At the request of the referee, variables that may have non-linear effects were added to the models on page 7. For Model 2, considering that the effect of vaccination on the case may be increasing, the square of the variable was also included in the model. However, there was a linear relationship between the number of cases and vaccination. In countries where Covid-19 cases are very high, the number of deaths is affected by herd immunity. Therefore, the square of the number of cases was added to model 3 as a variable. The results obtained showed that there is a non-linear relationship between the number of cases and covid-19 deaths.